# Correlation of Genotype-Phenotype of Congenital Hypothyroidism Cohort Diagnosed by Newborn Screening: A Long-Term Observational Study

**DOI:** 10.3390/ijns11040098

**Published:** 2025-10-20

**Authors:** Yajie Su, Xifeng Lei, Ayijiamali Muhetaer, Jinfeng He, Long Li

**Affiliations:** 1Department of Neonatology, Children’s Hospital of Xinjiang Uygur Autonomous Region, Xinjiang Hospital of Beijing Children’s Hospital, Urumqi 830054, China; yajiesu922@163.com (Y.S.); ayijiamali3268@126.com (A.M.); 2Department Graduate of School, Xinjiang Medical University, Urumqi 830054, China; 15059940141@139.com; 3Department of Neonatology, People’s Hospital of Xinjiang Uygur Autonomous Region, Urumqi 830054, China

**Keywords:** congenital hypothyroidism, gene spectrum, genotype–phenotype, drug maintenance dosage, outcomes

## Abstract

This long-term observational study aimed to define the spectrum of genetic variation in a congenital hypothyroidism (CH) cohort and investigate the correlations between specific genotypes and clinical phenotypes, including treatment requirements and outcomes. We analyzed the maintenance dose of L-thyroxine (L-T4) at 6, 12, 18, and 24 months, alongside clinical outcomes after 3 years. Data were collected from the Neonatal Disease Screening Center at our hospital between January 2011 and March 2024. Of 247 patients with confirmed CH, 119 had available genetic testing and complete clinical information. The genetic positivity rate was 56.3% (67/119). *DUOX2* was the most frequently mutated gene (28.57%), followed by *TPO*, *TG*, and *TSHR*. Phenotypic correlation analysis revealed that patients with *DUOX2* variants had significantly lower initial screening TSH levels and required lower L-T4 maintenance doses at 12 months compared to those with *TPO* or *TSHR* variants. Patients with *TPO* and *TSHR* variants exhibited more severe clinical phenotypes and a higher prevalence of thyroid enlargement on ultrasound. Notably, no significant differences in biochemical data, L-T4 doses, or clinical outcomes were observed between patients with monoallelic and biallelic *DUOX2* variations, or among the negative, monogenic, and oligogenic variation groups. This study establishes a high genetic diagnostic yield for CH in the studied cohort, with *DUOX2* as the predominant genetic etiology. The findings demonstrate significant genotype–phenotype correlations, where variations in different genes are associated with distinct biochemical severities and treatment demands. Crucially, the lack of correlation between the number of affected *DUOX2* alleles and disease severity highlights the complex genetic and phenotypic heterogeneity of CH. These results provide valuable insights for the precise management and prognostic counseling of patients with CH.

## 1. Introduction

Congenital hypothyroidism (CH) is a prevalent and preventable endocrine disorder worldwide, with an increasing annual incidence of primary CH now estimated at 1/3000–1/2000 [1]. CH arises from defects in thyroid gland development or hormone biosynthesis, traditionally classified as thyroid dysgenesis (TD) or dyshormonogenesis (DH) [2]. Patients with CH lack specific clinical symptoms in the early stage, such as enlarged fontanel, delayed regression of jaundice, umbilical hernia, abdominal distension, and constipation [3]. If untreated in early infancy, CH can result in irreversible mental and physical growth retardation [4,5,6].

Targeted genetic testing identified variants is up to 66.5% in patients with CH [7,8,9,10]. Over 20 genes are associated with CH, such as *TSHR*, *TTF-1*/*NKX2-1*, *PAX-8*, *TTF-2*/*NKX2-5*, *FOXE1*, *DUOX2*, *DUOXA2*, *TG*, *TPO*, *SLC5A5*/*NIS*, *SLC26A4*/*PDS*, and *IYD* [11]. The prevalence of specific gene variants varies among populations; *DUOX2* variants are common in Chinese cohorts, while *TPO* variants predominate in Caucasian populations [12]. However, available genetic and clinical data show a lack of genotype–phenotype correlation and significant phenotypic heterogeneity among patients with the same genotype. While CH is often considered as monogenic disorder, more studies and several pedigrees confirmed the potential oligogenic origin of CH [9,13]. The pathogenic contribution of these genes and their relationship with the clinical phenotype require further clarification [13].

This study conducted a comprehensive analysis of clinical phenotypes, maintenance L-T4 doses, clinical outcomes, and genotypes of infants diagnosed with CH via newborn screening over 13 years. The objectives were to establish the genetic variation spectrum of CH and elucidate genotype–phenotype relationships to facilitate precise patient management.

## 2. Materials and Methods

### 2.1. Study Design

A retrospective study was conducted to analyze the correlation of genotype–phenotype in children diagnosed with CH according to the Consensus Guidelines [14] at the Newborn Screening Center in our hospital between January 2011 and March 2024. A total of 119 patients with CH were enrolled in this study (Figure 1) and written informed consent was obtained from their parents for the collection of samples and publication of medical data. This study was approved by the Ethics Board of Children’s Hospital of Xinjiang Uygur Autonomous Region (KY2022031207).

### 2.2. Participants and Data Collection

The diagnosis of CH was made according to the Chinese national consensus statement and European guidelines. [1,14] A pretreatment blood thyrotropin stimulating hormone (TSH) concentration at newborn screening (NBS) of ≥ 9.5 mIU/L on initial screening was required. The diagnosis of CH was confirmed by elevated serum TSH levels and FT4 levels below the reference range.

Patients were further filtered based on the following exclusion criteria: (1) individuals lacking next-generation sequencing (NGS) data of CH-related genes, and (2) treatment with L-T4 was abandoned after diagnosis, and (3) clinical information was including pretreatment clinical presentation, thyroid ultrasonography, therapeutic dose of L-T4 at 6, 12, 18 and 24 months.

### 2.3. Laboratory Assessments

The screening TSH levels from dried blood spots were measured using a fluorometric assay with the Neonatal TSH kit (Fenghua, China) and an Auto Fluoroimmunoassay Analyzer (Auto TRFIA-2, Fenghua, China). Serum TSH and FT4 concentration were detected using a Roche electrochemiluminometric analyzer (Cobas-e601 analyzer, Roche, Mannheim, Germany).

### 2.4. Thyroid Ultrasonography Examination

Thyroid ultrasonography was performed on all patients using a 12 MHz small-parts linear transducer connected to an EnVisor scanner (Philips, Amsterdam, The Netherlands). The infants were placed in a supine position with their necks hyperextended without sedation. The length, breadth, and depth of the thyroid gland were measured, and the thyroid volume of each lobe was calculated as volume = length × breadth × depth × 0.479. The sum of the left and right thyroid lobe volumes was considered the total volume, excluding the isthmus. Gland volume was defined as enlarged, normal, or hypoplastic compared with published data for Chinese infants aged 0 to 12 months [15].

### 2.5. Targeted Next-Generation Sequencing and Variant Annotations

Peripheral blood samples were collected, and genomic DNA was extracted using the QIAamp DNA Blood Mini Kit (Qiagen, Hilden, Germany) according to the manufacturer’s instructions. DNA fragments were enriched for clinical exome sequencing using the ClearSeq Inherited Disease panel kit (Agilent Technologies, Santa Clara, CA, USA), which covered 2742 genes and included 21 genes related to CH. Sequencing was performed on a HiSeq 2500, HiSeq X10, or NovaSeq 6000 platform (Illumina, San Diego, CA, USA).

The pathogenicity of each variant was reassessed according to the ClinVar annotations https://www.ncbi.nlm.nih.gov/clinvar/ (accessed on 16 April 2025). Novel variants and variants of uncertain significance in the ClinVar were reassigned based on the guidelines of the ACMG [16].

### 2.6. Classification of Variants in Patients with CH

The genetic classification was defined into three groups based on the pattern of detected variants. Negative group: no pathogenic (P)/likely pathogenic (LP) variant was detected in CH-related genes; patients with variants of uncertain significance (VUS), benign or likely benign based on ACMG guideline were included here. Monogenic variation group: only one CH-related gene was detected and the P/LP variants were consistent with the mode of inheritance. The information on CH-related genes that we detected is shown in Appendix A. Oligogenic variation group: two or more CH-related genes were detected and the single P/LP heterozygous variant of one gene with autosomal recessive inherited mode was also included.

### 2.7. Definition of Phenotypic Indicators

Clinical presentation: the clinical manifestations at diagnosis (pretreatment), including jaundice, rough skin, constipation, etc. Severe CH was defined by a very low pretreatment serum FT4 < 5 pmol/L. Therapeutic dose of L-T4 (μg/kg·d) was recorded at the age of 6 ± 1 months, 12 ± 1 months, 18 ± 1 months and 24 ± 1 months. To determine clinical outcome, L-T4 therapy was temporarily withdrew after 3 years. Thyroid function was re-examined at 1 month, 2 months and 10 months, post-withdrawal. If TSH and FT4 levels remained normal, the patients were diagnosed as transient CH (TCH). If the results were abnormal, therapy was resumed, and the patient was diagnosed as permanent CH (PCH). Patients under 3 years of age were defined as “undiagnosed classification” [17].

### 2.8. Statistical Analyses

Statistical analysis was performed using Stata 16.0 software. Continuous variables were presented as mean ± standard deviation or median (Q1, Q3), while categorical variables were expressed as frequency and percentage. The normality of continuous variables was determined using the Shapiro–Wilk test. For comparisons among multiple groups, the Kruskal–Wallis test was used for continuous variables, and the chi-square or Fisher’s exact tests were used for categorical variables. If the Kruskal–Wallis test indicated a statistically significant difference (*p* < 0.05), post hoc pairwise comparisons were conducted using Dunn’s test with a Bonferroni correction for multiple comparisons. The Wilcoxon rank sum test was used to compare the intergroup differences for continuous variables. Group differences were evaluated using the chi-square or Fisher’s exact tests. Statistical significance was set at a *p* value less than 0.05. Graphs were created using GraphPad Prism 8 software.

## 3. Results

### 3.1. Genetic Spectrum of 119 Patients with CH by Next-Generation Sequencing

Genetic testing and complete clinical information were obtained for 119 out of 247 patients diagnosed with CH. In this cohort, 52 patients were classified into the negative group, as no P/LP variants in CH-related genes were identified. Monogenic variants consistent with Mendelian inheritance patterns were identified in 56 cases (47.06%, 56/119), which constituted the monogenic variants group. Meanwhile, 13 patients were found to carry P/LP variants in at least two CH-associated genes, which constituted the Oligogenic variants group. The genetic positivity rate was established at 56.3% (67/119) (Figure 1).

A total of 102 diverse variants were detected in the 14 genes examined in 119 patients, 82 were P/LP variants detected in the following genes: *DUOX2* in 34 patients (28.57%, 34/119), *TPO* in 20 patients (16.81%, 20/119), *TG* in 13 patients (10.92%, 13/119), *TSHR* in 11 patients (9.24%, 11/119), *DUOXA2*. *SLC26A4* and *SLC5A5* in 2 patients each (1.68%, 2/119), *GNAS*, *NKX2-1*, *PAX8*, *JAG1*, *NKX2-5* in one patient each (0.84%,1/119) (Appendix A).

### 3.2. Relationship Between Different Genes and Clinical Phenotypes

The four genes with the highest variation frequencies were selected to correlate their variation types with patients’ clinical phenotypes. For the *DUOX2* gene, missense variants were the most common variant (52.5%), among which 58.1% of patients presented with severe phenotypes (pretreatment serum FT4 < 5 pmol/L). Notably, only 26.7% of those with nonsense variants exhibited severe clinical manifestations (Figure 2A). For *TPO* gene, missense variants accounted for 68.4%, and 76.9% of these cases were severe phenotypes. Additionally, all patients with splice variants (5.3%) showed severe phenotypes (Figure 2B). For *TG* gene, missense variants predominated (70.3%), with 56.5% leading to severe phenotypes. Nonsense and frameshift variants each accounted for 13.5%, while splice variants constituted 2.7%; all patients with splice variants exhibited severe phenotypes (Figure 2C). For *TSHR* gene, missense variants were most common (81.3%), followed by frameshift (12.5%) and splice variations (6.3%). Except for 7.7% of patients with missense variants who had mild CH, all other patients presented with severe phenotypes (Figure 2D).

Each pie chart consists of two layers: the first is the distribution of variation types, with green indicating frameshift variants, blue for missense variants, purple for splice variants, and yellow for nonsense variants. The yellow-labeled numbers indicate the proportion of each variation type. The second shows the proportion of patients with severe clinical phenotypes, depicted in dark pink, with black numerals specifying the exact percentage. Panels A–D present *DUOX2*, *TPO*, *TG*, and *TSHR*, respectively.

Further analysis was performed on the baseline characteristics and clinical phenotypes associated with variations in the four genes. The Kruskal–Wallis test revealed a statistically significant difference in initial screening TSH levels across the four genetic variant groups (*p* = 0.022). Numerically, the median TSH level in the *DUOX2* group was lower than that in the *TPO* group, although post hoc testing with Bonferroni correction did not identify significant pairwise comparisons (100.0 (79.8, 120.0) vs. 172.1 (108.2, 221.6), *p* = 0.018). (Figure 3A). In addition, a statistically significant difference in the L-T4 dose was observed between the *DUOX2* and *TSHR* groups at 12 months 3.2 (1.2) vs. 4.5 (0.7), (*p* = 0.042) (Figure 4A). No significant correlation was observed between various genotypes and clinical outcomes, and PCH was the predominant outcome in our cohort. Patients with *TPO* and *TSHR* variation exhibited severe phenotypes compared to those with *DUOX2* variations (*DUOX2* vs. *TPO*: *p* = 0.011. *f DUOX2* vs. *TSHR*: *p* = 0.006). Similarly, a significantly higher prevalence of enlarged thyroid ultrasound was observed in patients with *TG* and *TPO* variants compared with *TSHR* variants. (*TG* vs. *TSHR*: *p* = 0.025. and *TPO* vs. *TSHR*: *p* = 0.011) (Table 1).

### 3.3. The Differences Between “DUOX2 Monoallelic Variations” and “DUOX2 Biallelic Variations”

The analysis focused on the most common *DUOX2* variations, involving 5 patients with monoallelic variations and 23 with biallelic variations. No significant differences were observed in the biochemical data, maintenance L-T4 doses at different ages, and clinical prognosis between the *DUOX2* monoallelic and *DUOX2* biallelic variation groups (Figure 3C,D and Figure 4B) (Appendix A).

A–B show the differences in the biochemical data between *DUOX2*, *TG*, *TPO*, and *TSHR* gene variations groups. C–D show the differences in the biochemical data between “*DUOX2* monoallelic variations” and “*DUOX2* biallelic variations” groups. E–F show the differences in the biochemical data between “Negative”, “Monogenic variations” and “Oligogenic variations” groups. The data are presented as the median with interquartile range, with each scatter point representing an individual patient; only statistically significant differences are indicated, using asterisks (*p* < 0.05).

### 3.4. The Differences Between “Negative”, “Monogenic Variations” and “Oligogenic Variations” Groups

For the between-group analysis, 52 patients were classified as negative 54 as monogenic variations, and 13 as oligogenic. The information on oligogenic variations patients is showed in Table 2. No significant differences in biochemical characteristics or L-T4 doses were found among the negative, monogenic, and oligogenic variation groups. However, the negative group was noted to have fewer cases of enlarged thyroid tissue at diagnosis and fewer cases of PCH after 3 years of age (Figure 3E,F and Figure 4C) (Appendix A).

## 4. Discussion

In this study, we screened for CH-related genes and genotype–phenotype relationships in 119 patients with CH. Biochemical data, thyroid morphology, drug maintenance dosages, and clinical prognosis were compared across different genetic groups. Previous studies indicated that while 85% of CH cases have historically been attributed to TD [18], an increased proportion of CH cases caused by DH has been reported recently [19,20].

In our cohort, the genetic positivity rate was 56.3%, with *DUOX2* being the most frequently mutated gene, a finding consistent with previous reports in East Asian populations [21,22]. However, *TPO* has been reported as the most frequently mutated genes in Western Caucasian populations [12].The high-frequency variants p.K530X and p.R1110Q were identified in the *DUOX2* gene, with the p.R1110Q variant having been previously demonstrated to significantly reduce H_2_O_2_ production in vitro [23] with p.K530X being the most common variation in southern and central Chinese populations also [7,24,25]. A study in Shandong Province, China, did not detect the p.K530X variant but identified the p.R1110Q variant as high-frequency [26]. In Korean populations, the common *DUOX2* variant is p.G488R [21], further highlighting genetic variations across diverse populations.

In the genotype–phenotype analysis, missense variants were identified as the most common type of variation among the four frequently occurring mutant genes. In the present study, *TSHR* variations mainly present with severe hypothyroidism, while *DUOX2* variations were less common. Zhang et al. [27] indicated that patients with *TSHR* variations resulted in milder hypothyroidism compared to *DUOX2* variations. The discrepancy in findings may be related to sample selection. Moreover, *TSHR* variations can lead to irreversible damage by causing receptor dysfunction, which disrupts thyroid gland development and interrupts the stimulating signal for hormone synthesis, thereby predisposing affected individuals to PCH.

Based on biochemical data and maintenance doses, patients with *DUOX2* variations generally exhibited milder clinical phenotypes than those with variations in other genes. The critical role of *TPO* in thyroid hormone synthesis, for which no functional substitute is known, may explain why *TPO* variations often lead to more severe symptoms and necessitate higher L-T4 doses [18,28]. Consistently, in our cohort, patients with *TPO* and *TSHR* variants were observed to exhibit more severe clinical phenotypes compared to the *DUOX2* group. Accordingly, the *DUOX2* group was found to require a lower L-T4 dosage at 12 months of age. We hypothesize that milder presentation in *DUOX2* variation patients could be attributed to partial functional compensation by *DUOX1*.

We observed phenotypic variations not only across different genes but also among different allelic variants of the same gene, such as *DUOX2*, an essential component of the thyroid H_2_O_2_ generation system, which plays a pivotal role in thyroid hormone production [29]. *DUOX2* variations, a primary cause of thyroid hormone production disorders, predominantly exhibit an autosomal recessive inheritance pattern. However, pathogenic variants can arise even from single allele variations, contributing to considerable genotype–phenotype variability. Moreno et al. [30] report that PCH is associated with biallelic inactivating *DUOX2* variations, while TCH is linked to monoallelic variations. Some studies suggest that monoallelic *DUOX2* variations typically correlate with mild to moderate phenotypes [24,31]. However, subsequent research suggests that the permanence or transience of CH is not directly linked to the number of *DUOX2* alleles, and the relationship between *DUOX2* genotype and CH phenotype remains unclear [32,33,34,35]. In our cohort, significant differences were not observed in biochemical data, L-T4 doses, and clinical outcomes between *DUOX2* monoallelic variations and biallelic variations. Nonetheless, we found that the L-T4 maintenance doses were generally higher in patients with *DUOX2* monoallelic variations than those with *DUOX2* biallelic variations. This further highlights the genetic and phenotypic heterogeneity of *DUOX2* variations, emphasizing the need for additional in vitro functional experiments to elucidate their relationship.

Considerable phenotypic variability was also noted among different allelic variations in the same gene [29]. While biallelic inactivating *DUOX2* variations have been associated with PCH and monoallelic variations with TCH in some studies, this relationship remains unclear [36] and further functional studies are necessary to validate these underlying mechanisms [3]. In our cohort, no significant differences in biochemical data, L-T4 doses, or clinical outcomes were found between patients with *DUOX2* monoallelic and biallelic variations, highlighting the genetic and phenotypic heterogeneity of *DUOX2* variations and underscoring the need for functional studies to elucidate these relationships.

Several limitations of this study are acknowledged. First, thyroid scintigraphy was performed on only a few patients, which may have affected the accurate classification of CH into TD or DH. Second, in the analysis of clinical outcomes, only definitive PCH and TCH diagnoses after re-evaluation were considered. It is possible that some patients classified as PCH in this study might successfully discontinue treatment at an older age.

## 5. Conclusions

We detected 102 different variants across 14 CH-related genes in neonatal CH cohort and elucidated genotype–phenotype relationships. We found a high genetic diagnosis rate of 56.3% among neonates with CH who underwent genetic testing, and *DUOX2* as the most frequently mutated gene. Genotype–phenotype analysis provides clinicians with valuable insights for the precise follow-up management of CH.

## Figures and Tables

**Figure 1 IJNS-11-00098-f001:**
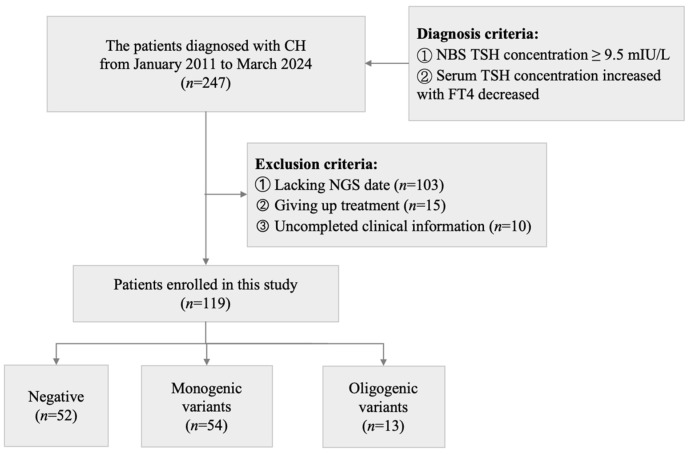
Study flow diagram. Abbreviations: CH, Congenital hypothyroidism; NBS, newborn screening; TSH, thyrotropin stimulating hormone; FT4, free thyroxine; NGS, next-generation sequencing.

**Figure 2 IJNS-11-00098-f002:**
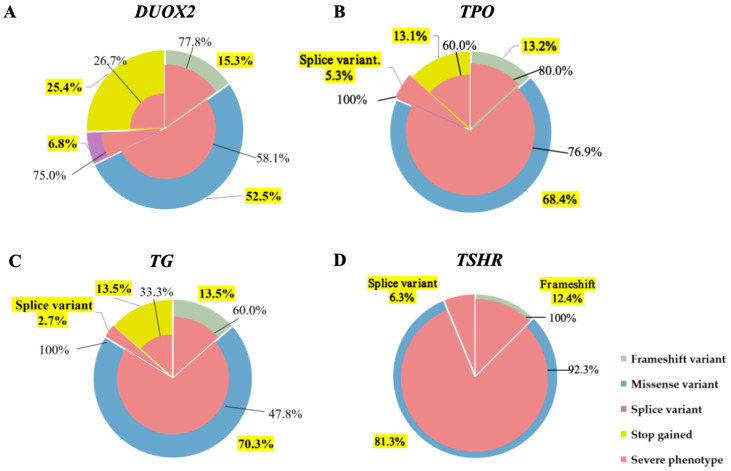
The correlation between different variation types of four high-frequency genes and the clinical phenotype of congenital hypothyroidism. (**A**) *DUOX2* variation with clinical phenotype. (**B**) *TPO* variation with clinical phenotype. (**C**) *TG* variation with clinical phenotype. (**D**) *TSHR* variation with clinical phenotype.

**Figure 3 IJNS-11-00098-f003:**
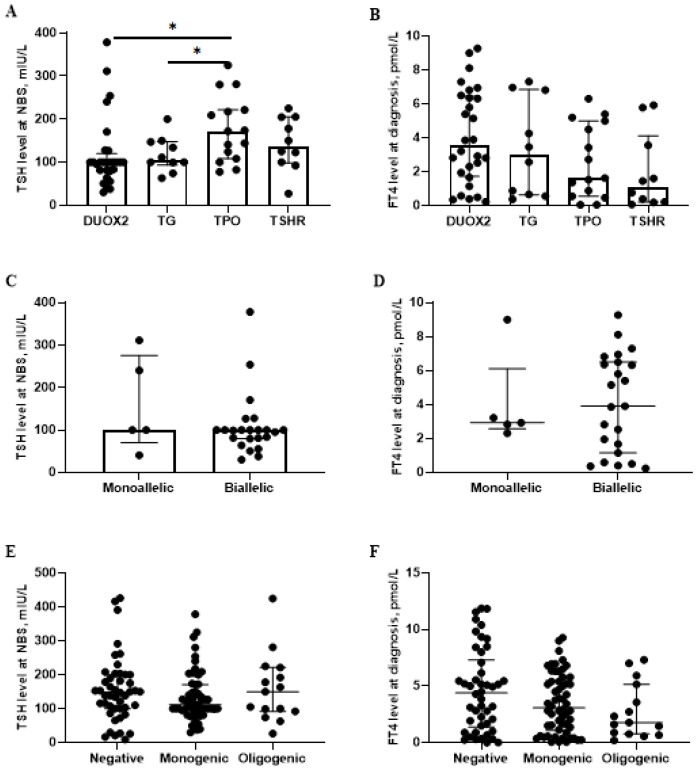
Genotype–phenotype of different genes and allelic in congenital hypothyroidism cohort. (**A**) TSH level at NBS of *DUOX2*, *TG*, *TPO* and *TSHR* variations groups. (**B**) FT4 level at diagnosis of *DUOX2*, *TG*, *TPO* and *TSHR* variations groups. (**C**) TSH level at NBS of “*DUOX2* monoallelic variations” and “*DUOX2* biallelic variations” groups. (**D**) FT4 level at diagnosis of “*DUOX2* monoallelic variations” and “*DUOX2* biallelic variations” groups. (**E**) TSH level at NBS of “Negative”, “Monogenic variations” and “Oligogenic variations” groups. (**F**) FT4 level at diagnosis of “Negative”, “Monogenic variations” and “Oligogenic variations” groups. “*” indicated a statistically significant difference (*p* < 0.05). Abbreviations: NBS, newborn screening; TSH, thyrotropin stimulating hormone; FT4, free thyroxine.

**Figure 4 IJNS-11-00098-f004:**
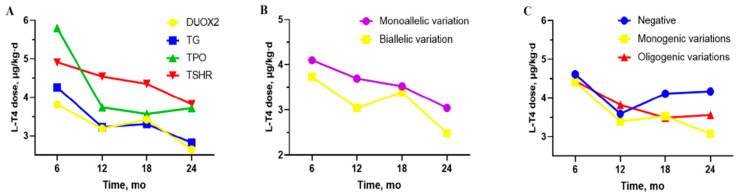
Drug maintenance dosages for different groups in infants aged 6–24 months. (**A**) L-T4 dose by age for *DUOX2*, *TG*, *TPO*, and *TSHR* variations groups. (**B**) L-T4 dose by age for “*DUOX2* monoallelic variations” and “*DUOX2* biallelic variations” groups. (**C**) L-T4 dose by age between “Negative”, “Monogenic variations” and “Oligogenic variations” groups.

**Table 1 IJNS-11-00098-t001:** Differences in the overall characteristics between *DUOX2*, *TG*, *TPO*, and *TSHR* variations groups.

Characteristic	*DUOX2* (*n* = 28)	*TG* (*n* = 10)	*TPO* (*n* = 15)	*TSHR* (*n* = 10)	*p* Value ^a^
Male *n* (%)	16 (57.1)	2 (20.0)	5 (33.3)	4 (40.0)	0.168
Ethnicity *n* (%)					
Han	16 (57.1) ^b^	2 (20.0)	1 (6.7)	1 (10.0)	0.001
Uygur	11 (39.3) ^c^	8 (80.0)	13 (86.7)	8 (80.0)	
TSH level at NBS, M (Q1, Q3), mIU/L	100.0 (79.8, 120.0)	105.5 (93.1, 147.8)	172.1 (108.2, 2 21.6) ^de^	137.5 (98.1, 205.0)	0.022
TSH level at diagnosis, M (Q1, Q3), mIU/L	100.0 (83.6, 100.0)	100.0 (91.2, 100.0)	100.0 (100.0, 150.0)	100.0 (100.0, 100.0)	0.096
FT4 level at diagnosis, M (Q1, Q3), pmol/L	3.5 (1.7, 6.5)	3.0 (0.7, 6.9)	1.64 (0.6, 5.0)	1.1 (0.2, 4.1)	0.089
L-T4 dose at 6 mo, mean (SD), μg/kg·d	3.8 (1.9)	4.3 (2.0)	5.8 (2.2)	4.9 (1.9)	0.059
L-T4 dose at 12 mo, mean (SD), μg/kg·d	3.2 (1.2)	3.2 (1.8)	3.7 (0.9)	4.5 (0.7) ^f^	0.049
L-T4 dose at 18 mo, mean (SD), μg/kg·d	3.4 (2.2)	3.3 (1.7)	3.6 (1.2)	4.4 (0.6)	0.458
L-T4 dose at 24 mo, mean (SD), μg/kg·d	2.6 (1.5)	2.8 (1.7)	3.7 (1.4)	3.8 (1.2)	0.220
Sever Clinical phenotype *n* (%)	11 (39.3)	3 (30.0)	12 (80.0) ^g^	9 (90.0) ^h^	0.003
Thyroid ultrasound *n* (%)					0.017
Hypoplasia	1 (3.6)	2 (20.0)	1 (6.7)	4 (40)	
Normal	20 (71.4)	4 (40.0)	7 (46.7)	6 (60)	
Enlarged	7 (25.0)	4 (40.0)	7 (46.7)	0 (0) ^ij^	
Outcome *n* (%)					0.528
PCH	21 (75.0)	9 (90.0)	14 (93.3)	9 (90.0)	
TCH	4 (14.3)	1 (10.0)	0 (0)	0 (0)	
Undiagnosed	3 (10.7)	0 (0)	1 (6.7)	1 (10.0)	

Abbreviations: TSH, thyroid stimulating hormone; NBS, newborn screening; FT4, free thyroxine; L-T4, levothyroxine; PCH, permanent congenital hypothyroidism; TCH, transient congenital hypothyroidism. ^a^
*p* value from χ^2^ test for categorical variables and Kruskal–Wallis test for continuous variables; unknown values were excluded. Significant pairwise comparisons after Dunn test with Bonferroni correction are indicated below the overall *p* values. ^b^
*DUOX2* vs. other groups *p* < 0.05. ^c^
*DUOX2* vs. other groups *p* < 0.05. ^d^
*DUOX2* vs. *TPO*: *p* = 0.018. ^e^
*TG* vs. *TPO*: *p* = 0.035. ^f^
*DUOX2* vs. *TSHR*: *p* = 0.042. ^g^
*DUOX2* vs. *TPO*: *p* = 0.011. ^h^
*DUOX2* vs. *TSHR*: *p* = 0.006. ^i^
*TG* vs. *TSHR*: *p* = 0.025. ^j^
*TPO* vs. *TSHR*: *p* = 0.011.

**Table 2 IJNS-11-00098-t002:** The genotypes and phenotypes information for 13 patients with congenital hypothyroidism. caused by oligogenic variations.

Case	Sex	Gene	Variation 1	Variation 2	TSH Level at NBS, mIU/L	Ultrasound	Clinical Outcome
8	F	*TPO*	c.940C > T	c.940C > T	281	Normal	PCH
		*NKX2-5*	c.641_642insGCC				
		*DUOX2*	c.2182G > A				
20	F	*TSHR*	c.1736C > T	c.1736C > T	225	Hypoplasia	PCH
		*DUOX2*	c.1295G > A				
26	F	*SLC26A4*	c.1790T > A		27	Hypoplasia	PCH
		*TSHR*	c.154C > A				
27	M	*TSHR*	c.394G > C		178	Hypoplasia	PCH
		*GNAS*	c.946G > C				
31	F	*TPO*	c.2677G > A		150	Normal	PCH
		*TSHR*	c.1349G > A				
38	F	*TG*	c.7198A > G	c.7198A > G	74.3	Hypoplasia	PCH
		*NKX2-1*	c.1054G > A				
39	F	*TSHR*	c.243-1G > C		92.31	Normal	PCH
		*TG*	c.4859C > T				
52	F	*DUOX2*	c.3175C > T		163.7	Hypoplasia	PCH
		*TPO*	c.256G > A				
60	M	*TSHR*	c.1942A > T	c.350T > G	100	Normal	PCH
		*DUOX2*	c.835G > A				
63	M	*DUOX2*	c.2654G > T		105.2	Enlarged	PCH
		*PAX8*	c.1046C > A				
66	M	*DUOX2*	c.1588A > T		192.05	Enlarged	TCH
		*DUOXA2*	c.412-412delinsTA				
79	F	*TPO*	c.1781G > A	c.1781G > A	221.61	NA	PCH
		*TPO*	c.1905G > C	c.1905G > C			
		*JAG1*	c.574T > C				
		*TSHR*	c.154C > T				
114	F	*SLC26A4*	c.1003T > C		424.91	Hypoplasia	NA
		*TPO*	c.2306G > A				

Abbreviations: M, male; F, female; NA, not available; PCH, permanent congenital hypothyroidism; TCH, transient congenital hypothyroidism.

## Data Availability

All data of this manuscript are available in Appendix A.

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
