# Peer review of "Correlation of Genotype-Phenotype of Congenital Hypothyroidism Cohort Diagnosed by Newborn Screening: A Long-Term Observational Study"

_2409-515X, 2025, doi:10.3390/ijns11040098_

Round 1

Reviewer 1 Report

Comments and Suggestions for Authors

The study aims to describe the genotype of a cohort of patients with congenital hypothyroidism identified through neonatal screening and to explore specific correlations between genotype and clinical phenotype. While the study provides interesting data regarding the genotypic characterization of a Chinese population with congenital hypothyroidism, some results appear inconsistent and imprecise, and thus require revision.

  • In the cohort description, it is not specified whether only patients with a eutopic thyroid were included, or whether cases with thyroid dysgenesis (ectopy, agenesia) were also considered. This distinction should be explicitly reported, especially given that the NGS analysis identified variants in genes associated with TD. This clarification is also fundamental if we consider the different pathogenetic mechanisms underlying the two forms of IC.
  • There are numerical inconsistencies in the Results. For instance, the abstract reports 81 pathogenic and likely pathogenic variants, whereas the results section states 82. Similarly, the number of patients in the monogenic group is unclear: Figure 1 (flow chart) indicates 54 patients, while section 3.4 of the Results reports 58. The authors are therefore requested to carefully revise and ensure consistency across all reported data.
  • The Results state that one patient carried mutations in TBL1X gene,  associated with central hypothyroidism. Were patients with central forms of hypothyroidism also included in the study?
  • Considering that therapeutic reassessment is carried out between the ages of 2 and 3 years, it might be more appropriate to include patients also < 3 years of age in the 'undefined/undiagnosed' group.
  • Given the wide variability between genotype and phenotype already described in the literature for CH, and considering the role that the combination of individual variants may have in determining the clinical presentation, it would be interesting to present the complete genotype for each patient, either in the results or as supplementary material.
  • Regarding TSHR, the authors mention an autosomal dominant inheritance pattern. How many patients with TSHR variants had monoallelic or biallelic mutations?
  • Since an extended genotype description is not provided, it also remains unclear how many patients in the monogenic group were homozygous or compound heterozygous for autosomal recessive genes. In cases of monogenic autosomal recessive inheritance, was parental testing performed to confirm inheritance of the variants in trans? Are there also clinical data from the parents available?
  • 7 out of 15 patients in the “oligogenic” group carried two variants in the same gene, in addition to single heterozygous variants in different genes. Do the authors believe that this finding may have influenced the results, particularly about the clinical characteristics of this group?
  • Not all relevant findings are adequately discussed. For example, how do the authors explain the high percentage of permanent forms among patients with TSHR mutations, which are generally associated with variable phenotypes and often non-severe forms of congenital hypothyroidism?

Author Response

The study aims to describe the genotype of a cohort of patients with congenital hypothyroidism identified through neonatal screening and to explore specific correlations between genotype and clinical phenotype. While the study provides interesting data regarding the genotypic characterization of a Chinese population with congenital hypothyroidism, some results appear inconsistent and imprecise, and thus require revision.

Comment 1: In the cohort description, it is not specified whether only patients with a eutopic thyroid were included, or whether cases with thyroid dysgenesis (ectopy, agenesia) were also considered. This distinction should be explicitly reported, especially given that the NGS analysis identified variants in genes associated with TD. This clarification is also fundamental if we consider the different pathogenetic mechanisms underlying the two forms of IC.

Response: Thank you for this comment. We agree that distinguishing between eutopic thyroid and thyroid dysgenesis is critical. Our study cohort included patients with both conditions: those with a eutopic thyroid gland and those with thyroid dysgenesis (including ectopy and agenesis). We have now explicitly stated this in the revised manuscript (Line 77-78).

As the reviewer rightly points out, the pathogenic mechanisms differ. However, we included patients with variants in thyroid dysgenesis-associated genes regardless of ultrasound findings because a normal ultrasound does not definitively rule out such variants. The primary aim of our genetic analysis was to identify the molecular etiology, which can be independent of the structural phenotype. The data presented in this study allow us to investigate potential associations between specific genetic variants and dyshormonogenesis.

Comment 2: There are numerical inconsistencies in the Results. For instance, the abstract reports 81 pathogenic and likely pathogenic variants, whereas the results section states 82. Similarly, the number of patients in the monogenic group is unclear: Figure 1 (flow chart) indicates 54 patients, while section 3.4 of the Results reports 58. The authors are therefore requested to carefully revise and ensure consistency across all reported data.

Response: Thank you for pointing out the basic mistakes. The number of 81 in the abstract indicated patients and 82 in the results indicated variants.Similarly, the number of 54 in Figure 1 indicated patients and 58 in the results indicated variants.  We have revised and uniformed the concept in the full text.

Comment 3: The Results state that one patient carried mutations in TBL1X gene, associated with central hypothyroidism. Were patients with central forms of hypothyroidism also included in the study?

Response:Thank you for this pertinent observation. Our cohort was specifically selected based on the diagnostic criteria for primary congenital hypothyroidism. We did not include patients with central hypothyroidism. The single case with a TBL1X variant was categorized within the group of “Patients carried variants of VUS/B/LB based on ACMG guideline” due to the classification of VUS. Considering that the previous division into four groups might mislead readers, in the revised manuscript, we reclassified and defined VUS and B/LB variants that did not conform to the genetic pattern as negative, and reanalyzed the data.

Comment 4: Considering that therapeutic reassessment is carried out between the ages of 2 and 3 years, it might be more appropriate to include patients also < 3 years of age in the 'undefined/undiagnosed' group.

Response:Thank you. We corrected the time for clinical assessment to after the age of 3 years and rechecked the patient's clinical results at this moment. The total count of undiagnosed patients itself remains unchanged from our original data.

Comment 5: Given the wide variability between genotype and phenotype already described in the literature for CH, and considering the role that the combination of individual variants may have in determining the clinical presentation, it would be interesting to present the complete genotype for each patient, either in the results or as supplementary material.

Response:Thank you. The genotypic information for each patient has been provided in the Supplementary TableS4.

Comment 6: Regarding TSHR, the authors mention an autosomal dominant inheritance pattern. How many patients with TSHR variants had monoallelic or biallelic mutations?

Since an extended genotype description is not provided, it also remains unclear how many patients in the monogenic group were homozygous or compound heterozygous for autosomal recessive genes. In cases of monogenic autosomal recessive inheritance, was parental testing performed to confirm inheritance of the variants in trans? Are there also clinical data from the parents available?

Response:Thank you. In the Supplementary TableS4, we have provided detailed genetic information for all 119 patients. Among them, 11 patients carried pathogenic TSHR variants. One patient had a monoallelic (heterozygous) variant, which was confirmed by parental testing to be a de novo variant. Five patients carried either compound heterozygous or homozygous. The remaining six patients detected TSHR variants along with pathogenic variants in other CH-related genes, and were divided into Oligogenic variants group.

Comment 7: 7 out of 15 patients in the “oligogenic” group carried two variants in the same gene, in addition to single heterozygous variants in different genes. Do the authors believe that this finding may have influenced the results, particularly about the clinical characteristics of this group?

Response:Thank you for raising this important point. In the revised manuscript, we have reclassified patients with variants of uncertain significance (VUS) into the negative group. Following this adjustment, only two patients (Cases 64 and 79) were found to carry four pathogenic variants in the same gene (TPO). We believe it was contributed to a more severe clinical presentation. Both infants exhibited markedly elevated TSH levels on newborn screening and have since been confirmed to have permanent congenital hypothyroidism. We described it in discussion.

Comment 8: Not all relevant findings are adequately discussed. For example, how do the authors explain the high percentage of permanent forms among patients with TSHR mutations, which are generally associated with variable phenotypes and often non-severe forms of congenital hypothyroidism?

Response:Thank you. We have now expanded the discussion to address the notably high rate of permanent congenital hypothyroidism among patients with TSHR variants in our cohort. Line 261-264.

Reviewer 2 Report

Comments and Suggestions for Authors

Summary

This manuscript describes the assessment of genotype-phenotype correlations in a large cohort of babies with congenital hypothyroidism (CH) seen at one hospital in China.  Among 247 babies with CH who were screen-positive for CH and confirmed by low serum free T4 and high serum TSH, the authors had CH-related genetic results and complete clinical information for 119 of them.  Of those, 69 were found to have known pathologic variants.   The 119 babies were categorized by their genetic testing results and compared regarding their screening TSH, diagnostic TSH, diagnostic free T4 and l-thyroxine dose at 6, 12, 18 and 24 months.  A few significant correlations were identified.

Strengths

The authors present an unusually large cohort of babies with CH who underwent genetic testing and they thoroughly examined the relationship of those results to clinical findings.

Using the whole cohort of 247, the least possible fraction that had a genetic explanation for their CH was 28%.  As the authors note, this is much higher than other reports and may be due to the genetic background of their patient population.

Areas for Improvement

I think the statement in the Abstract that “pathogenic mutations in CH-related genes were identified in 81 cases of 119 patients” suggests a much higher incidence of positive genetic testing than was actually present.  A clarifying sentence probably needs to precede that one stating that “Genetic testing and complete clinical information were available in 119 of 247 babies with confirmed CH.”  Similarly, I suspect that the citation of a 60% mutation rate (line 46) is taking the other studies out of context and that not all patients in those studies had genetic testing.

I think the data analysis in Table 1 may need to be redone.  For example, I don’t think it is relevant that among the Han patients, the DUOX2 predominated.  What would be relevant is comparing the distribution of genetic diagnoses between the Han and Uygur patients, which would result in a single p value for the two lines in the Table.  Similarly, I think thyroid ultrasound and outcomes need to be “correlated” as a group with genetic diagnoses.  In the same Table, I worry that calculating a mean TSH may not be valid.  TSH levels are usually right-skewed and not normally distributed.  Stating the median and interquartile range may be more appropriate.  Comparing across genetic diagnoses would require using a non-parametric test.  The same concern applies to all the parts of Figure 2 that report TSH levels.

Section 3.1 states that DUOX2 mutations were found in 36 patients, but section 3.2 reports on only 28, with similar discrepancies for the other genotypes.  The Methods need to state why only a subset of patients were included in the between-group analysis.

Minor Points

The first “sentence” of the Abstract is not a sentence.

I do not understand why there are two Supplementary files rather than incorporating all the tables into one file.

Line 135 states “14 genes and 102 diverse variants were detected.”  I think the authors mean, “102 diverse variants were detected in the 14 genes examined.”

In figure 2A, it is not valid to report a p value comparing 2 of the 4 groups without including a Bonferroni correction for having done multiple inter-group comparisons.  The same concern applies to Figure 3A’s comparison of DUOX2 and TSHR groups.  I assume the asterisk implies a low p value, but could not find that stated in the Figure legend.

In Figure 2E, I do not understand why the statistic is reported as “NA” rather than a p value.

Comments on the Quality of English Language

There are several places, especially in the Discussion, where the wording is not consistent with English idiom.  Examples below are not all of the instances:

line 206: accounted for the vast majority proportion

line 208: which likely due to genetic background variations

line 214: the DUOX2 mutations p.K530X and p.R1110Q were high frequency of variants

line 216: demonstrated that the variants was markedly reduced H2O2 216 production activity

line 241: mutation cases predominantly showed normal struc-241 tures or enlarged

Author Response

This manuscript describes the assessment of genotype-phenotype correlations in a large cohort of babies with congenital hypothyroidism (CH) seen at one hospital in China. Among 247 babies with CH who were screen-positive for CH and confirmed by low serum free T4 and high serum TSH, the authors had CH-related genetic results and complete clinical information for 119 of them. Of those, 69 were found to have known pathologic variants.  The 119 babies were categorized by their genetic testing results and compared regarding their screening TSH, diagnostic TSH, diagnostic free T4 and l-thyroxine dose at 6, 12, 18 and 24 months. A few significant correlations were identified.

Strengths

The authors present an unusually large cohort of babies with CH who underwent genetic testing and they thoroughly examined the relationship of those results to clinical findings.

Using the whole cohort of 247, the least possible fraction that had a genetic explanation for their CH was 28%. As the authors note, this is much higher than other reports and may be due to the genetic background of their patient population.

Areas for Improvement

Comment 1:I think the statement in the Abstract that “pathogenic mutations in CH-related genes were identified in 81 cases of 119 patients” suggests a much higher incidence of positive genetic testing than was actually present. A clarifying sentence probably needs to precede that one stating that “Genetic testing and complete clinical information were available in 119 of 247 babies with confirmed CH.” Similarly, I suspect that the citation of a 60% mutation rate (line 46) is taking the other studies out of context and that not all patients in those studies had genetic testing.

Response:Thank you. We agree that the original statement could be misinterpreted and have revised the manuscript to provide clearer context. In the Abstract, we have now preceded the sentence with the following clarification: “Genetic testing and complete clinical information were available for 119 of the 247 patients with confirmed CH.”(Line 20-21). Furthermore, we have re-examined the cited references (studies 7-10) and acknowledge that their reported high diagnostic yields also refer to targeted genetic investigations in selected cohorts. Accordingly, we have modified the statement in the introduction (Line 44) to:“Some studies report that the identified variants is as high as 66.5% in patients with CH who undergo targeted genetic testing.”

Comment 2:I think the data analysis in Table 1 may need to be redone. For example, I don’t think it is relevant that among the Han patients, the DUOX2 predominated. What would be relevant is comparing the distribution of genetic diagnoses between the Han and Uygur patients, which would result in a single p value for the two lines in the Table. Similarly, I think thyroid ultrasound and outcomes need to be “correlated” as a group with genetic diagnoses.

Response: We sincerely thank the comment. We completely agree that the initial statistical approach for the categorical variables was not optimal. As suggested, we have now re-analyzed the data for the Ethnicity, Thyroid ultrasound, and Clinical outcome variables in table 1.

Comment 3:In the same Table, I worry that calculating a mean TSH may not be valid. TSH levels are usually right-skewed and not normally distributed. Stating the median and interquartile range may be more appropriate. Comparing across genetic diagnoses would require using a non-parametric test. The same concern applies to all the parts of Figure 2 that report TSH levels.

Response: Thank you for this important methodological insight. We agree that TSH levels are typically not normally distributed. In the revised manuscript, we have recalculated and now present both initial TSH and FT4 levels as the median with interquartile range in the table 1. Accordingly, the statistical comparisons of hormone levels across genetic diagnostic groups in Figure 3 have been updated using non-parametric tests.

Comment 4:Section 3.1 states that DUOX2 mutations were found in 36 patients, but section 3.2 reports on only 28, with similar discrepancies for the other genotypes. The Methods need to state why only a subset of patients were included in the between-group analysis.

Response: Thank you for highlighting this inconsistency. In our original analysis, we had classified patients with variants of VUS into a separate category that was not included in the comparative analysis between genotype-positive and genotype-negative groups. Considering that the previous division into four groups might mislead readers, in the revised manuscript, we reclassified and defined VUS and B/LB variants that did not conform to the genetic pattern as negative, and reanalyzed the data. Consequently, all patient data are now included in a unified analysis, and the numbers reported throughout the manuscript (including in Sections 3.1 and 3.2) are consistent and refer to the same, complete cohort.

Minor Points

1.The first “sentence” of the Abstract is not a sentence.

2.I do not understand why there are two Supplementary files rather than incorporating all the tables into one file.

3.Line 135 states “14 genes and 102 diverse variants were detected.” I think the authors mean, “102 diverse variants were detected in the 14 genes examined.”

4.In figure 2A, it is not valid to report a p value comparing 2 of the 4 groups without including a Bonferroni correction for having done multiple inter-group comparisons. The same concern applies to Figure 3A’s comparison of DUOX2 and TSHR groups. I assume the asterisk implies a low p value, but could not find that stated in the Figure legend.

5.In Figure 2E, I do not understand why the statistic is reported as “NA” rather than a p value.

Response: Thank you for these detailed and constructive comments. We have carefully addressed each point in the revised manuscript and For Figures 2, "NA" or “*” has been replaced with the appropriate notation

Comments on the Quality of English Language

There are several places, especially in the Discussion, where the wording is not consistent with English idiom. Examples below are not all of the instances:

line 206: accounted for the vast majority proportion

line 208: which likely due to genetic background variations

line 214: the DUOX2 mutations p.K530X and p.R1110Q were high frequency of variants

line 216: demonstrated that the variants was markedly reduced H2O2 216 production activity

line 241: mutation cases predominantly showed normal struc-241 tures or enlarged

Response: Thank you. We have carefully reviewed the entire manuscript, with particular attention to the Discussion section, and have thoroughly polished the language to improve clarity, consistency, and adherence to standard English academic style.

Round 2

Reviewer 2 Report

Comments and Suggestions for Authors

Summary

This manuscript describes the assessment of genotype-phenotype correlations in a large cohort of babies with congenital hypothyroidism (CH) seen at one hospital in China.  A few significant correlations were identified.

Strengths

The authors did a good job of responding to the reviewers’ comments.

Areas for improvement

Line 237 states, “In the present cohort, the vast majority of patients exhibited enlarged thyroid tissue or a gland-in-situ, suggesting that DH is the primary pathophysiology of CH in this Chinese population, a finding likely attributable to genetic background differences.”  I think that is a misleading, circular statement because the cohort was probably selected for genetic testing because of having an enlarged or in-situ gland.  I think the sentence should be deleted.

As stated in my original review, it is not valid to report a p value comparing two categories when there are 4 categories being examined.  The appropriate Kruskal-Wallis p value appears in Table 1 and should be cited in the text rather than the difference between just two categories.  Once a statistically significant difference has been shown among all the categories, individual pairs of categories may be compared using the Bonferroni correction.  Almost all of the paragraph that starts on line 176 should be revised.  This also applies to line 262 and possibly other parts of the Discussion.

Minor points

The Kruskal-Wallis test is appropriately used to evaluate the data in Table 1, but is not mentioned in the Methods.

Tables S2 and S3 need to change “mutation” to “variation”.

line 299 – “of 56.3% among neonates with CH” should be “of 56.3% among neonates with CH who underwent genetic testing”.  Without the qualification, the 56% will be cited and misinterpreted as applying to all infants with CH.

Comments on the Quality of English Language

There are problems with subject/verb agreement and plurals.  See highlights in attachment.  Some highlights refer to content and not English usage.  Those are discussed in Comments and Suggestions.

Author Response

This manuscript describes the assessment of genotype-phenotype correlations in a large cohort of babies with congenital hypothyroidism (CH) seen at one hospital in China.  A few significant correlations were identified.

Strengths

The authors did a good job of responding to the reviewers’ comments.

Areas for improvement

Comment 1:Line 237 states, “In the present cohort, the vast majority of patients exhibited enlarged thyroid tissue or a gland-in-situ, suggesting that DH is the primary pathophysiology of CH in this Chinese population, a finding likely attributable to genetic background differences.”  I think that is a misleading, circular statement because the cohort was probably selected for genetic testing because of having an enlarged or in-situ gland.  I think the sentence should be deleted.

Response:Thank you for this critical insight. We agree that the statement was potentially misleading, as the cohort undergoing genetic testing was indeed enriched for patients with abnormal thyroid imaging, which does not support a generalization about the broader Chinese CH population. We have therefore deleted the sentence

Comment 2: As stated in my original review, it is not valid to report a p value comparing two categories when there are 4 categories being examined.  The appropriate Kruskal-Wallis p value appears in Table 1 and should be cited in the text rather than the difference between just two categories.  Once a statistically significant difference has been shown among all the categories, individual pairs of categories may be compared using the Bonferroni correction.  Almost all of the paragraph that starts on line 176 should be revised.  This also applies to line 262 and possibly other parts of the Discussion.

Response: Thank you again for this methodological correction. We agree that the previous approach of reporting direct pairwise comparisons without adjustment for multiple testing was statistically inappropriate. We have now re-analyzed the data as follows: For variables involving comparisons across the four genetic groups (DUOX2, TG, TPO, TSHR), we first confirmed a statistically significant overall difference using the Kruskal-Wallis test (as originally reported in Table 1). For pairs showing a significant overall difference, we proceeded with Dunn's post-hoc test for pairwise comparisons, applying the Bonferroni correction to control the family-wise error rate. In the reversed Table 1, the superscripts b to g were respectively marked, and the p-values after pairwise comparisons were listed in the table notes. We reversed the text starting from line 176.

Minor points

The Kruskal-Wallis test is appropriately used to evaluate the data in Table 1, but is not mentioned in the Methods.

Tables S2 and S3 need to change “mutation” to “variation”.

line 299 – “of 56.3% among neonates with CH” should be “of 56.3% among neonates with CH who underwent genetic testing”.  Without the qualification, the 56% will be cited and misinterpreted as applying to all infants with CH.

There are problems with subject/verb agreement and plurals.  See highlights in attachment.  Some highlights refer to content and not English usage.  Those are discussed in Comments and Suggestions.

Response: Thank you very much for your careful review of the full text. We have checked the grammar and subject-verb agreement throughout the manuscript.